# LTP and memory impairment caused by extracellular Aβ and Tau oligomers is APP-dependent

Daniela Puzzo[1], Roberto Piacentini[2], Mauro Fá[3], Walter Gulisano[1], Domenica D Li Puma[2], Agnes Staniszewski[3], Hong Zhang[3], Maria Rosaria Tropea[1], Sara Cocco[2], Agostino Palmeri[1], Paul Fraser[4], Luciano D'Adamio[5], Claudio Grassi[2], Ottavio Arancio[6]*

[1]Department of Biomedical and Biotechnological Sciences, University of Catania, Catania, Italy; [2]Institute of Human Physiology, Università Cattolica del Sacro Cuore, Rome, Italy; [3]Department of Pathology and Cell Biology and Taub Institute for Research on Alzheimer's Disease and the Aging Brain, Columbia University, New York, United States; [4]Tanz Centre for Research in Neurodegenerative Diseases and Department of Medical Biophysics, University of Toronto, Toronto, Canada; [5]Department of Microbiology and Immunology, Albert Einstein College of Medicine, New York, United States; [6]Department of Pathology and Cell Biology and Taub Institute for Research on Alzheimer's Disease and the Aging Brain, Columbia University, New york, United States

**Abstract** The concurrent application of subtoxic doses of soluble oligomeric forms of human amyloid-beta (oAβ) and Tau (oTau) proteins impairs memory and its electrophysiological surrogate long-term potentiation (LTP), effects that may be mediated by intra-neuronal oligomers uptake. Intrigued by these findings, we investigated whether oAβ and oTau share a common mechanism when they impair memory and LTP in mice. We found that as already shown for oAβ, also oTau can bind to amyloid precursor protein (APP). Moreover, efficient intra-neuronal uptake of oAβ and oTau requires expression of APP. Finally, the toxic effect of both extracellular oAβ and oTau on memory and LTP is dependent upon APP since APP-KO mice were resistant to oAβ- and oTau-induced defects in spatial/associative memory and LTP. Thus, APP might serve as a common therapeutic target against Alzheimer's Disease (AD) and a host of other neurodegenerative diseases characterized by abnormal levels of Aβ and/or Tau.

*For correspondence: oa1@columbia.edu

Competing interests: The authors declare that no competing interests exist.

## Introduction

Protein aggregation and deposition have been considered key pathogenetic processes in several neurodegenerative disorders, including Alzheimer's Disease (AD), tauopathies, Parkinson's Disease, Huntington disease and many others (*Shelkovnikova et al., 2012*; *Takalo et al., 2013*). More recently, soluble small aggregates of these proteins have gained a lot of attention in studies aimed at understanding the etiopathogenesis of these diseases. This is particularly evident in AD, in which the abnormal increases of the levels of amyloid-beta (Aβ) and Tau proteins and their aggregation are crucial steps in the chain of events leading to dementia (*Irvine et al., 2008*; *Kopeikina et al., 2012*).

The importance of soluble oligomeric forms of Aβ (oAβ) and Tau (oTau) has been corroborated by numerous evidences demonstrating their presence in human cerebrospinal fluid in healthy

individuals and, in higher amounts, in AD patients (*Hölttä et al., 2013*; *Sengupta et al., 2017*). oAβ and oTau are also toxic to cell-to-cell communication, as they disrupt synaptic plasticity, paving the way to the subsequent cognitive impairment (*Selkoe, 2008*; *Lasagna-Reeves et al., 2012*; *Fá et al., 2016*). Interestingly, we have recently demonstrated that a brief exposure to a combination of sub-toxic doses of extracellular oAβ and oTau dramatically reduces memory and its electrophysiological surrogate long-term potentiation (LTP) (*Fá et al., 2016*). These findings beg the question of whether they act through a common pathway when they impair memory and LTP.

Aβ and Tau share numerous common biochemical features. Both proteins can form insoluble deposits: that is, extracellular amyloid plaques due to the accumulation of Aβ, and intracellular insoluble filaments and neurofibrillary tangles formed by Tau. In addition, Aβ and Tau are present as non-fibrillar soluble monomeric and oligomeric species (*Selkoe, 2008*; *Lasagna-Reeves et al., 2010*; *Fá et al., 2016*). They can be secreted at the synapse in an activity-dependent fashion (*Kamenetz et al., 2003*; *Pooler et al., 2013*; *Yamada et al., 2014*; *Fá et al., 2016*), and enter neurons (*Frost et al., 2009*; *Lai and McLaurin, 2010*; *Wu et al. 2013*; *Fá et al., 2016*). Moreover, both Aβ and Tau can bind to amyloid precursor protein (APP) (*Lorenzo et al., 2000*; *Van Nostrand et al., 2002*; *Shaked et al., 2006*; *Fogel et al., 2014*; *Takahashi et al., 2015*), a protein with a central role in AD pathogenesis that might act as a cell surface receptor (*Deyts et al., 2016*).

APP, the precursor of Aβ, which derives from sequential cleavage of APP by β-secretase (also known as BACE1) and γ-secretase (*Cole and Vassar, 2007*; *De Strooper, 2007*), has a central role in AD pathogenesis and might act as both an Aβ precursor and a cell surface receptor (*Deyts et al., 2016*). Here we have postulated that oAβ and oTau involve APP as a common mechanism of action when they impair memory and LTP. This has been investigated through a series of experiments in which we have used APP knock-out (APP-KO) mice and assayed whether suppression of APP function blocks the deleterious effects of both oAβ and oTau onto memory and LTP.

## Results

### Similar to oAβ, oTau binds to APP

APP has been shown to bind both Aβ and Tau (*Lorenzo et al., 2000*; *Van Nostrand et al., 2002*; *Shaked et al., 2006*; *Fogel et al., 2014*; *Takahashi et al., 2015*). The interaction between oAβ and APP has been thoroughly investigated in studies demonstrating that different species of Aβ (monomers, dimers, oligomers and fibrils) bind to APP (*Lorenzo et al., 2000*; *Van Nostrand et al., 2002*; *Shaked et al., 2006*; *Fogel et al., 2014*). However, there is no proof that oTau binds to APP, as previous studies on Tau-APP binding did not use oligomers but fibrils (*Giaccone et al., 1996*; *Islam and Levy, 1997*; *Smith et al., 1995*; *Takahashi et al., 2015*). We therefore decided to investigate whether the interaction between Tau and APP can be extended to oTau. This was accomplished through two different approaches. In the first one, we utilized membrane fractions from HEK293 cells stably transfected with human APP with the Swedish mutation (APPSw) and incubated with/out oTau derived from recombinant 4R/2N Tau protein. After incubation APP was immuno-precipitated (IP) and the IPs were tested for oTau binding. As shown in *Figure 1A*, APP co-IPed oTau. In the second approach, as an alternative method to analyze protein-protein interaction dependent upon the presence of endogenous APP, we performed far-WB (fWB) on cultured hippocampal neurons from either wild type (WT) or APP-KO animals. We found that, in lysates from WT cultures, oTau (used as the bait protein) was detected at the molecular weight of APP (~110 KDa) by an anti-Tau antibody (Tau 5). Conversely, this band was not observed in lysates from control APP-KO cultures (*Figure 1B*), supporting the interaction between murine APP and oTau. Collectively, these experiments demonstrate that oTau is able to bind APP.

### Expression of APP is required for efficient intra-neuronal uptake of oAβ and oTau

The similarity between Aβ and Tau can be extended to the entrance of their oligomers into neurons from the extracellular space (*Frost et al., 2009*; *Lai and McLaurin, 2010*; *Wu et al. 2013*; *Fá et al., 2016*). Given that both Aβ and Tau can bind to APP, our next goal was to establish whether APP is needed for oligomer internalization. To address this issue, we treated cultured hippocampal neurons obtained from WT and APP-KO mice with either 200 nM oAβ labeled with HiLyte Fluor 555

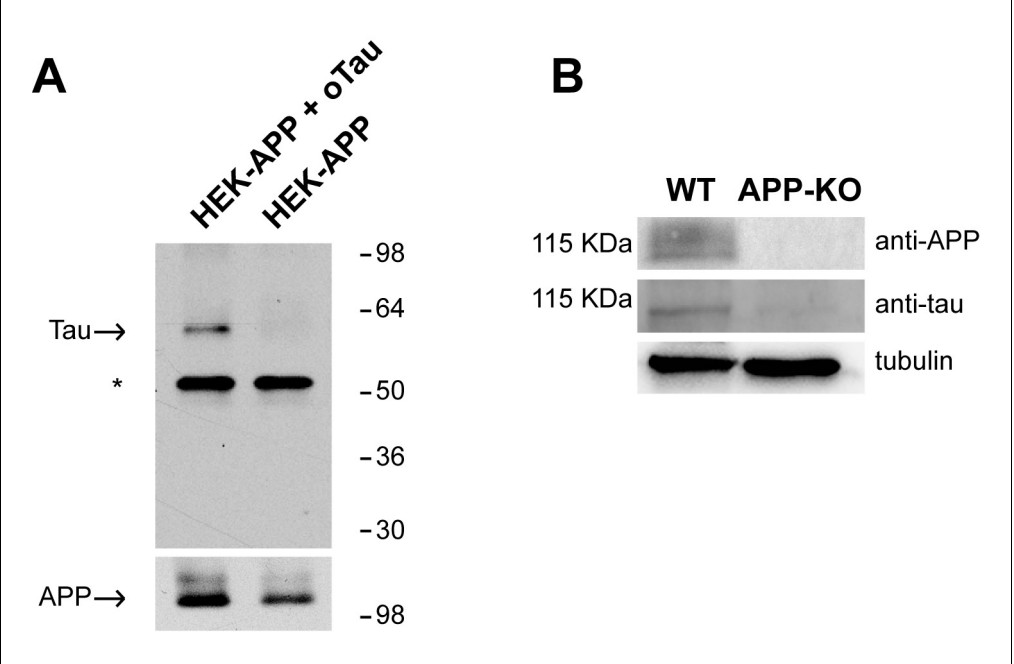

**Figure 1.** APP binds to oTau. (**A**) WB with anti-Tau antibodies Tau5 showing oTau co-IPed with APP in HEK293 cells stably transfected with human APP with the Swedish mutation. * corresponds to the heavy chain of the antibody used for IP. (**B**) Representative data from fWB experiments performed on hippocampal neurons from WT and APP-KO mice, showing interaction between APP and Tau. Tau binding to APP is demonstrated by the presence of bands recognized by Tau5 antibodies at 110 KDa (the molecular weight of APP). Tubulin was used as loading control. n = 3.

(oA$\beta$−555) or 100 nM oTau labeled with IRIS-5 ester dye (oTau-IRIS5) for 20 min and we studied their cellular internalization by high-resolution confocal microscopy using an automated algorithm to detect and count intraneuronal spots. We found that WT neurons internalized much more A$\beta$ and Tau than APP-KO cells. In fact, after extracellular oA$\beta$−555 application, a higher percentage of WT neurons exhibited A$\beta$ accumulation compared to APP-KO cultures (*Figure 2A*). A$\beta$ accumulation was found in 91 ± 3% of WT MAP2-positive cells (*Figure 2B*, *Figure 2—source data 1*) and the mean number of intracellular fluorescent spots/neuron was 5.3 ± 0.4 (*Figure 2C*, *Figure 2—source data 1*). When the same treatment was applied to APP-KO cultures we found that 73 ± 5% of total cells internalized A$\beta$ (*Figure 2B*, *Figure 2—source data 1*) and the mean number of fluorescent spots was 2.9 ± 0.2 (*Figure 2C*, *Figure 2—source data 1*). Similar results were obtained when WT and APP-KO neurons were treated with extracellular oTau-IRIS5 (*Figure 2D*) which was found in 80 ± 6% of WT cells containing 2.7 ± 0.2 fluorescent spots and in 47 ± 6% of APP-KO neurons exhibiting 1.4 ± 0.1 spots (*Figure 2E–F*, *Figure 2—source data 2*). Moreover, to provide a global estimate of the protein uploading into neurons, we performed quantitative analysis of these data through the 'internalization index', which showed a 61% reduction in APP-KO neurons compared to WT cells for oA$\beta$ (*Figure 2G*, *Figure 2—source data 3*), and a 69% reduction for oTau (*Figure 2H*, *Figure 2—source data 3*). Notably, the amounts of A$\beta$ and tau oligomers attached to neuronal surface did not significantly differ between WT and APP-KO cells. Specifically, fluorescent A$\beta$ spots were 6.9 ± 0.5 and 6.5 ± 0.6 for WT and APP-KO, respectively (*Figure 2I*, *Figure 2—source data 4*); whereas for Tau they were 4.3 ± 0.4 and 4.0 ± 0.4, respectively (*Figure 2J*, *Figure 2—source data 4*). Collectively, these data show that APP suppression reduces the entrance of extracellular oligomers of both A$\beta$ and Tau into neurons.

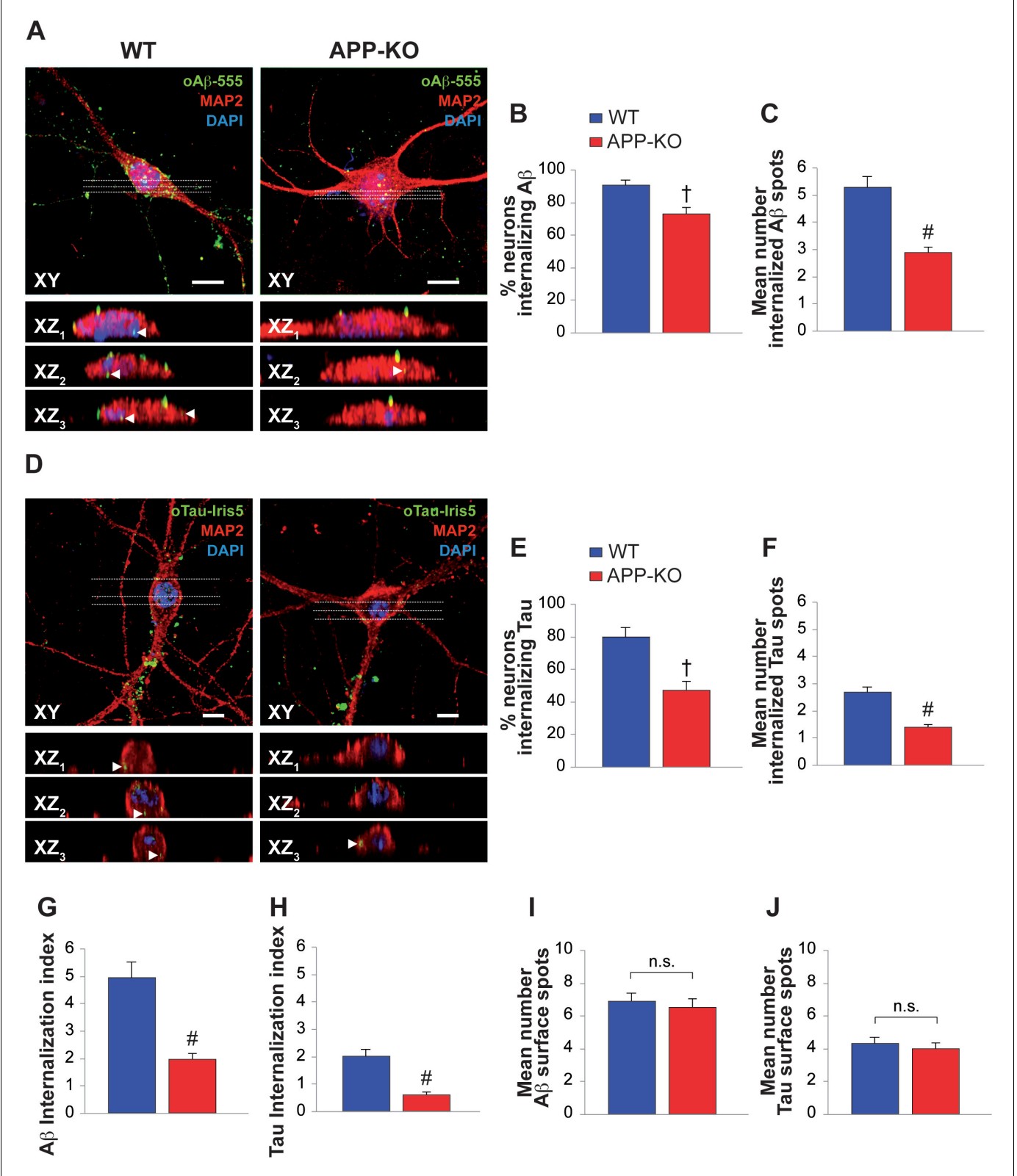

**Figure 2.** APP suppression reduces internalization of oAβ and oTau into neurons. (**A**) Representative images of cultured hippocampal neurons (microtubule associated protein-2 (MAP2) positive cells) obtained from either WT or APP-KO mice and treated with 200 nM human oligomeric Aβ42 labeled with HiLyteTM Fluor 555 (oAβ−555) for 20 min and immunostained for MAP-2. Lower images show different XZ cross-sections from the acquired confocal Z-stack referring to the dotted lines numbered as 1–3 in each panel. Arrowheads indicate internalized proteins. Scale bars: 10 μm. (**B–**
*Figure 2 continued on next page*

*Figure 2 continued*

**C**) After 20 min of extracellular oAβ−555 application, the percentage of WT neurons exhibiting Aβ accumulation was 91 ± 3% of total cells (n = 127) and the mean number of intracellular fluorescent spots/neuron was 5.3 ± 0.4. When the same treatment was applied to APP-KO cultures we found that 73 ± 5% of total cells internalized Aβ (n = 112; *t* test: $t_{(98)}$ = 2.734; p=0.007 comparing APP-KO vs. WT cells) and a markedly lower mean number of fluorescent spots (2.9 ± 0.2; $t_{(191)}$ = 4.508; p<0.0001 comparing APP-KO vs. WT cells). (**D**) Representative images of WT and APP-KO cultured hippocampal neurons treated with 100 nM IRIS-5-labeled human recombinant oligomeric Tau (oTau-IRIS5) for 20 min. Lower images show different XZ cross-sections from the acquired confocal Z-stack referring to the dotted lines numbered as 1–3 in each panel. Arrowheads indicate internalized proteins. Scale bars: 10 μm. (**E–F**) After 20 min of extracellular Tau-IRIS5, the percentage of WT neurons exhibiting Tau was 80 ± 6% of WT cells (n = 88) with 2.7 ± 0.2 fluorescent spots, whereas 47 ± 6% of APP-KO neurons showed Tau internalization (n = 84; $t_{(71)}$ = 3.945; p=0.0002) with a mean number of fluorescent spots equal to 1.4 ± 0.1 ($t_{(92)}$ = 4.331; p<0.0001). (**G–H**) The 'internalization index' shown on the graph was 4.9 ± 0.6 in WT neurons treated with Aβ−555 vs. 1.9 ± 0.2 of APP-KO cells ($t_{(98)}$ = 5.246; p<0.0001), and 2.0 ± 0.3 in WT neurons treated with Tau-IRIS5 vs. 0.6 ± 0.1 of APP-KO cells ($t_{(71)}$ = 5.013; p<0.0001). (**I**) Fluorescent Aβ spots attached to neuronal surface were 6.9 ± 0.5 and 6.5 ± 0.6 for WT and APP-KO, respectively ($t_{(170)}$ = 0.576; p=0.56). (**J**) Fluorescent Tau spots attached to neuronal surface were 4.3 ± 0.4 and 4.0 ± 0.4 for WT and APP-KO, respectively ($t_{(93)}$ = 0.363; p=0.72).

The following source data is available for figure 2:

**Source data 1.** Data relating to *Figure 2B–C*.
**Source data 2.** Data relating to *Figure 2E–F*.
**Source data 3.** Data relating to *Figure 2G–H*.
**Source data 4.** Data relating to *Figure 2I–J*.

## The effect of extracellular oAβ onto memory depends upon the presence of endogenous APP

Neuronal uploading of oAβ from the extracellular space reduces LTP (*Ripoli et al., 2014*), a cellular surrogate of memory. Interestingly, both associative fear memory and spatial memory, two types of memory that are altered in AD patients, are impaired by oAβ (*Puzzo et al., 2014*). Thus, these effects may require intra-neuronal uptake of oAβ. Since APP is required for efficient uptake of oAβ, we evaluated the effect of oAβ onto two types of memory, assessed through Fear Conditioning and 2 day Radial Arm Water Maze (RAWM), respectively, in the presence or absence of functional APP expression using 3–4 month-old WT and APP-KO mice. Consisting with previous results (*Fiorito et al., 2013*; *Watterson et al., 2013*), high doses of oAβ (200 nM in a final volume of 1 μl, one injection 20 min prior to the training) infused via bilateral cannulas into the dorsal mouse hippocampi, resulted in reduced freezing 24 hr after the electric shock in WT mice (*Figure 3A*, *Figure 3—source data 1*), confirming that contextual fear memory is altered by high amounts of oAβ. By contrast, in interleaved experiments, memory was spared by the deleterious effects of oAβ in APP-KO mice (*Figure 3A*, *Figure 3—source data 1*). Similarly, APP-KO mice that were infused with vehicle displayed normal memory, as previously shown in KO animals of this age (*Senechal et al., 2008*) (*Figure 3A*, *Figure 3—source data 1*). We also confirmed that the defect in contextual memory found in WT mice was due to an oAβ-induced hippocampal impairment, whereas cued fear learning, a type of learning depending upon amygdala function (*Phillips and LeDoux, 1992*), was not affected in both WT and APP-KO animals treated with vehicle or oAβ (*Figure 3B*, *Figure 3—source data 1*). Moreover, we excluded that the defect was due to deficits in mouse capability to perceive the electric shock, as sensory threshold assessment did not reveal any difference among the four groups of mice (*Figure 3C*, *Figure 3—source data 1*).

We then evaluated short-term spatial memory with the RAWM. As previously shown (*Watterson et al., 2013*), WT mice infused with oAβ (200 nM in a final volume of 1 μl, one injection 20 min prior to the first trial of RAWM training in day one and two, bilaterally) made a higher number of errors than vehicle-infused WT littermates during the second day of RAWM testing (*Figure 3D*, *Figure 3—source data 2*). By contrast, the performance of APP-KO mice, which was normal when these animals were infused with vehicle, was not affected by the Aβ infusion (*Figure 3D*, *Figure 3—source data 2*). Control trials with a visible platform did not show any difference in speed or latency to reach the platform among the four groups, indicating that oAβ infusion did not affect the motility, vision and motivation of mice during RAWM testing (*Figure 3E–F*,

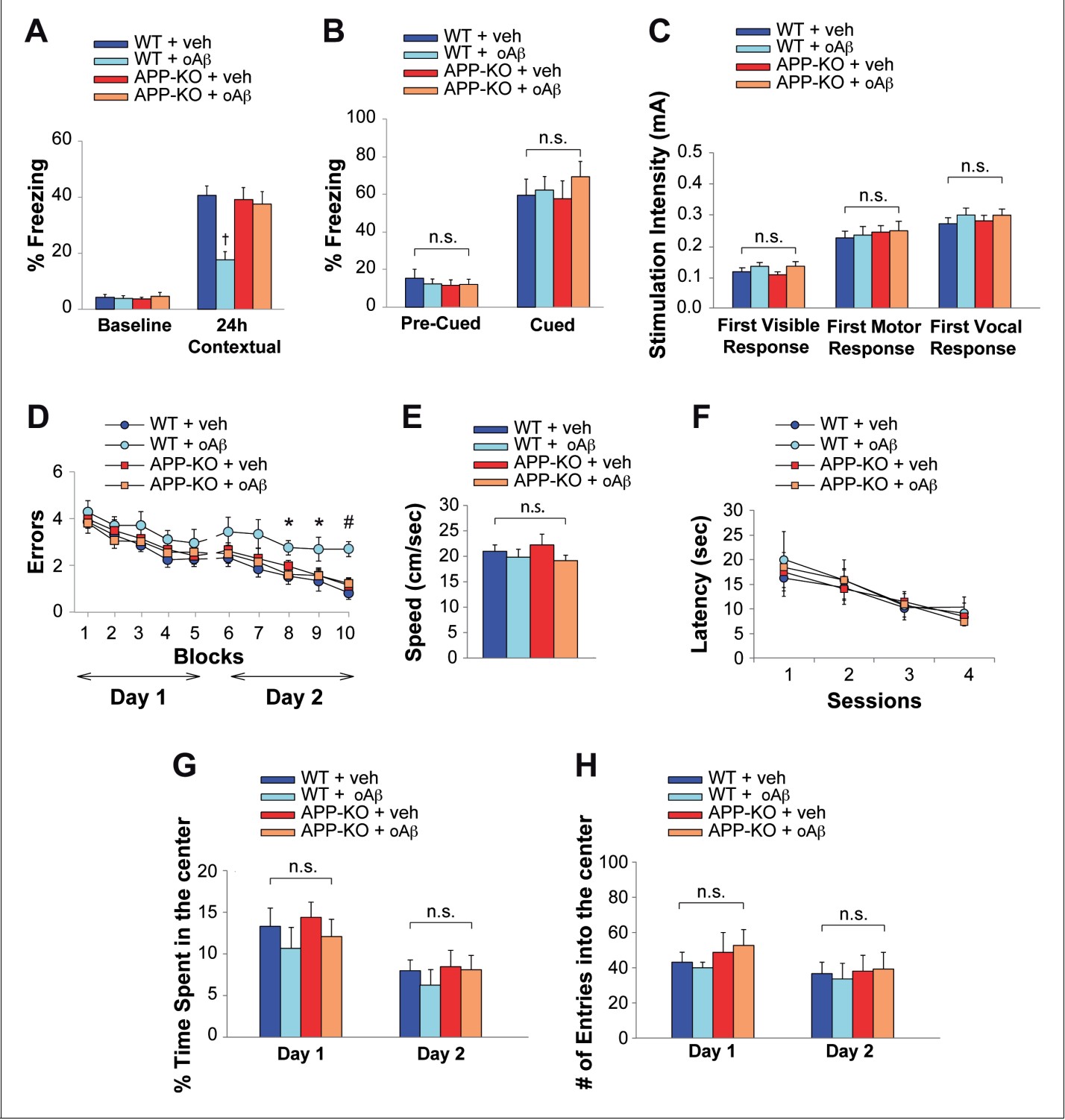

**Figure 3.** APP is necessary for extracellular oAβ to reduce memory. (**A**) oAβ (200 nM) impaired contextual memory in WT mice, whereas it did not impair memory in APP-KO mice. n = 11 per condition in this and the following panels. 24 hr: ANOVA $F_{(3,40)}$ = 8.047, p<0.0001; Bonferroni: WT + vehicle vs. WT + oAβ: † p<0.001. (**B**) Freezing responses before (Pre) and after (Post) the auditory cue were the same among vehicle- and oAβ-infused APP-KO mice as well as vehicle- and oAβ-infused WT littermates in the cued conditioning test. ANOVA Pre-Cued: $F_{(3,40)}$ = 0.242, p=0.867; Cued: $F_{(3,40)}$ = 0.372, p=0.774. (**C**) No difference was detected among the four groups during assessment of the sensory threshold. ANOVA for repeated measures $F_{(3,40)}$ = 0.626, p=0.602. (**D**) oAβ (200 nM) impaired the RAWM performance in WT mice whereas it did not impair the performance in APP-KO mice. ANOVA for repeated measures (day 2) $F_{(3,40)}$ = 5.297, p=0.004. WT + vehicle vs. WT + oAβ: *p<0.05 for block 8 and 9, and # p<0.0001 for block 10. (**E–F**) Testing

*Figure 3 continued on next page*

*Figure 3 continued*

with the visible platform task for assessment of visual-motor-motivational deficits did not reveal any difference in average speed [ANOVA: $F_{(3,40)}$ = 0.899, p=0.450] (E), and time to reach the visible platform [ANOVA for repeated measures $F_{(3,40)}$ = 0.05, p=0.985] (F) among the four groups. (G–H) Open field testing showed a similar percentage of time spent in the center compartment [ANOVA for repeated measures $F_{(3,40)}$ = 0.692 p=0.489] (G) and the number of entries into the center compartment [ANOVA for repeated measures $F_{(3,40)}$ = 0.332, p=0.802] (H) in vehicle- and oAβ-infused APP-KO mice as well as vehicle- and oAβ-infused WT littermates, indicating that they had no differences in exploratory behavior.

The following source data is available for figure 3:

**Source data 1.** Data relating to *Figure 3A–B–C*.
**Source data 2.** Data relating to *Figure 3D–E–F*.
**Source data 3.** Data relating to *Figure 3G–H*.

*Figure 3—source data 2*). Moreover, open field testing did not reveal any difference among WT and APP-KO mice treated with vehicle or oAβ, indicating that mouse exploratory behavior, which might affect animal performance in the memory task, was not affected by treatment or genotype (*Figure 3G–H*, *Figure 3—source data 3*). Collectively, these experiments indicate that the deleterious effect exerted by oAβ on memory is dependent upon the presence of endogenous APP.

## The effect of extracellular oTau onto memory depends upon the presence of endogenous APP

Both associative fear memory and spatial memory are impaired not only by oAβ, but also by oTau (*Fá et al., 2016*). As shown before, oTau binds APP and needs APP for an efficient entrance into neurons, just like oAβ. Thus, we tested if, similar to oAβ, exogenous oTau requires APP to alter memory. As previously demonstrated (*Fá et al., 2016*), oTau infusion (500 nM in a final volume of 1 µl, two injections bilaterally at 180 and 20 min prior to the electric shock for fear conditioning or the first trial of the RAWM training in day one and two) affected the two forms of memory in WT animals (*Figure 4A*, *Figure 4—source data 1* and *Figure 4D*, *Figure 4—source data 2*). By contrast, APP-KO mice displayed normal performance when they were infused with oTau both in the fear conditioning and RAWM tests (*Figure 4A*, *Figure 4—source data 1* and *Figure 4D*, *Figure 4—source data 2*). Moreover, we did not observe any behavioral differences between groups of mice tested for cued conditioning (*Figure 4B*, *Figure 4—source data 1*), sensory threshold (Figure, *Figure 4—source data 1*), visible platform (*Figure 4E and F*, *Figure 4—source data 2*) or open field (*Figure 4G and H*, *Figure 4—source data 3*). Thus, as for oAβ, the impairment of memory induced by oTau was dependent upon the presence of APP.

## APP is necessary for extracellular oAβ and oTau to reduce LTP

LTP represents a cellular correlate of learning and memory (*Bliss and Collingridge, 1993*). It is reduced after treatment with both high amounts of oAβ and/or oTau (*Fá et al., 2016*). Hence, we checked whether APP is needed for oAβ and oTau to impair LTP at the CA3-CA1 synapses. Following recording of basal synaptic transmission (BST), which did not reveal any difference between WT and APP-KO slices (*Figure 5A*, *Figure 5—source data 1*), slices were perfused with oAβ, or oTau, or vehicle prior to eliciting LTP through a theta-burst stimulation. As previously demonstrated (*Puzzo et al., 2005*), perfusion with oAβ (200 nM for 20 min before the tetanus) reduced LTP in slices from WT mice (*Figure 5B*, *Figure 5—source data 2*). However, consistent with the behavioral results, the peptide did not impair LTP in slices from APP-KO littermates (*Figure 5B*, *Figure 5—source data 2*). Similarly, oTau (100 nM for 20 min before tetanus) reduced LTP in WT slices but not in APP-KO slices (*Figure 5C*, *Figure 5—source data 3*).

Next, we checked whether the amyloidogenic processing of APP is required for oAβ and oTau toxicity. This was determined by using mice deficient in BACE1 (*Luo et al., 2001*). In previous WB analysis of these mice we had confirmed that they do not express BACE1 protein and have impaired β-processing of APP (*Del Prete et al., 2014*). BST recording did not reveal any difference between WT and BACE1-KO slices (*Figure 5D*, *Figure 5—source data 4*). Slices perfusion with oAβ (200 nM

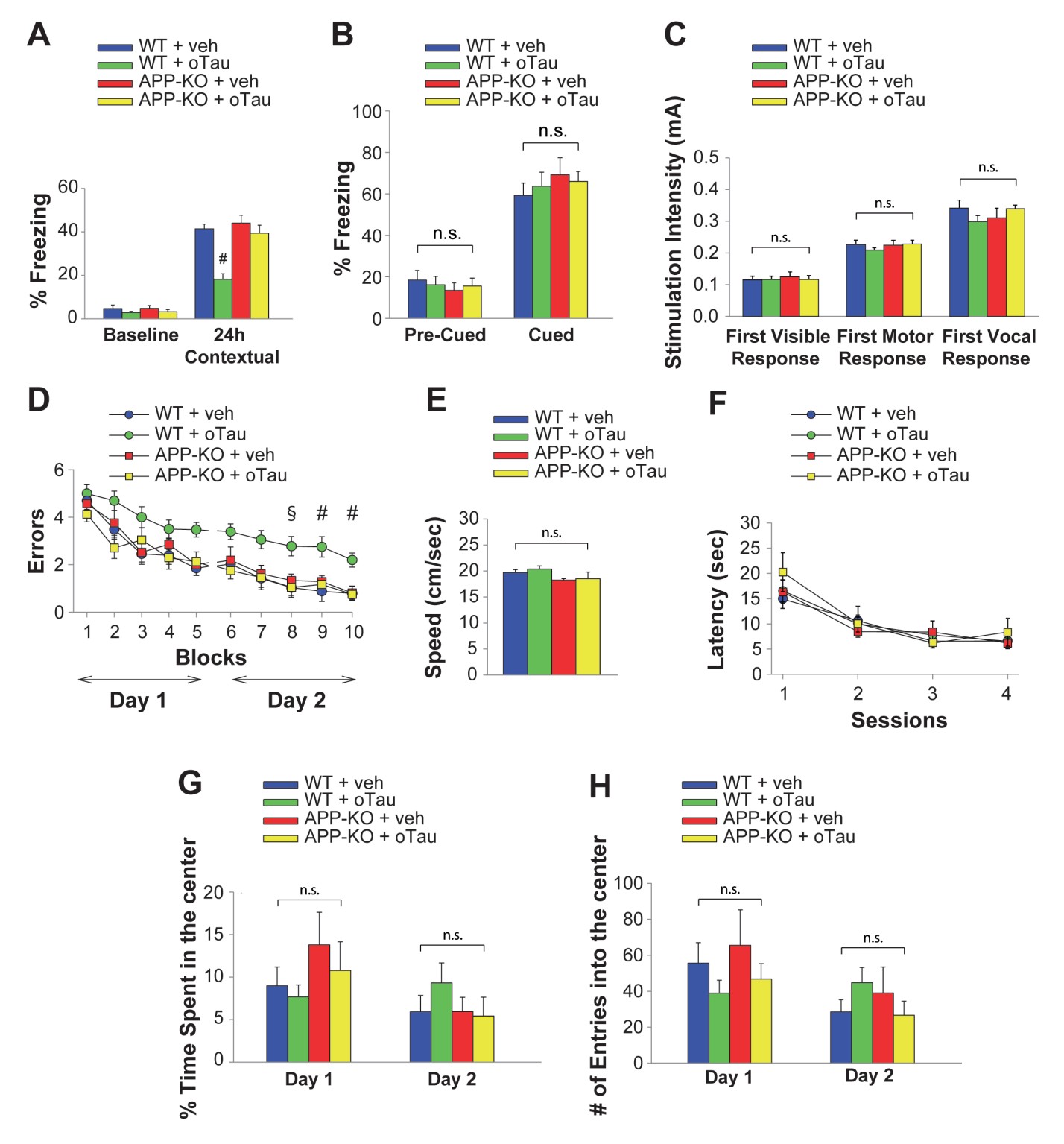

**Figure 4.** APP is necessary for extracellular oTau to reduce memory. (**A**) oTau (500 nM) impaired contextual memory in WT mice, whereas it did not impair contextual memory in APP-KO mice. 24 hr: ANOVA $F_{(3,38)}$ = 18.472, p<0.0001; Bonferroni: WT + vehicle vs. WT + oTau: # p<0.0001. WT + vehicle: n = 11, WT + oTau: n = 12, APP-KO + vehicle: n = 8, APP-KO + oTau: n = 11. (**B**) Freezing responses before (Pre) and after (Post) the auditory cue were the same among the four groups shown in A in the cued conditioning test. ANOVA Pre-cued: $F_{(3,38)}$ = 0.215, p=0.885; Cued: $F_{(3,38)}$ = 0.410, p=0.747. (**C**) No difference was detected among the four groups shown in A during assessment of the sensory threshold. ANOVA for repeated measures $F_{(3,38)}$ = 0.643, p=0.592. (**D**) oTau (500 nM) impaired the RAWM performance in WT mice whereas it did not impair the performance in APP-

*Figure 4 continued on next page*

*Figure 4 continued*

KO mice. ANOVA for repeated measures (day 2) $F_{(3,34)}$ = 11.309, p<0.0001. WT + vehicle vs. WT + oTau: § p<0.005 for block 8, and # p<0.001 for block 9 and 10. WT + vehicle: n = 11, WT + oTau: n = 12, APP-KO + vehicle: n = 7, APP-KO + oTau: n = 8. (E–F) Testing with the visible platform task for assessment of visual-motor-motivational deficits for animals shown in D did not reveal any difference in average speed [ANOVA: $F_{(3,34)}$ = 1.532, p=0.224] (E) and time to reach the visible platform [ANOVA for repeated measures: $F_{(3,34)}$ = 0.221, p=0.881] (F) among the four groups. (G–H) Open field testing for the same animals as in D showed a similar percentage of time spent in the center compartment [ANOVA for repeated measures $F_{(3,34)}$ = 0.190, p=0.902] (G) and the number of entries into the center compartment [ANOVA for repeated measures $F_{(3,34)}$ = 0.354, p=0.787] (H) in vehicle- and oTau-infused APP-KO mice as well as vehicle- and oTau-infused WT littermates, indicating that they had no differences in exploratory behavior.

The following source data is available for figure 4:

**Source data 1.** Data relating to *Figure 4A–B–C*.
**Source data 2.** Data relating to *Figure 4D–E–F*.
**Source data 3.** Data relating to *Figure 4G–H*.

for 20 min before the tetanus), or oTau (100 nM for 20 min before tetanus), or vehicle showed that, similar to WT mice, oAβ and oTau reduced LTP in slices from BACE1-KO mice (*Figure 5E*, *Figure 5—source data 5* and *Figure 5F*, *Figure 5—source data 6*). Thus, these experiments demonstrate that APP processing is not involved in the toxicity of extracellularly-applied Aβ and Tau.

Finally, we asked whether the APP-dependence for the negative effects of oAβ and oTau onto LTP is specific to these oligomers, or a broader property of APP with β-sheet, oligomer forming proteins. To address this question, we selected human amylin (Amy), an amyloid protein of 37 amino-acids differing from Aβ$_{42}$ in its primary sequence, but sharing with it the ability to form β-sheets and oligomerize (*Wineman-Fisher et al., 2016*). Amy crosses the blood brain barrier (*Banks et al., 1995*), and has a profile of neurotoxicity that is strikingly similar to that of Aβ (*Jhamandas et al., 2011*), including the marked reduction of LTP (*Kimura et al., 2012*). As previously demonstrated (*Kimura et al., 2012*), perfusion of hippocampal slices for 20 min with 200 nM oligomeric Amy (oAmy) produced a marked reduction of LTP in WT slices (*Figure 5G*, *Figure 5—source data 7*). The same impairment of LTP was observed in slices from APP-KO mice (*Figure 5G*, *Figure 5—source data 7*); thus, different than oTau and oAβ, oAmy does not require APP for its negative effect on synaptic plasticity. Collectively, these experiments suggest that a brief exposure to both oAβ or oTau, but not oAmy, needs the presence of endogenous APP to impair LTP.

## Discussion

Protein aggregate accumulation in the brain is a common feature to neurodegenerative diseases, each disease displaying specific aggregating proteins and aggregate distribution. Oligomers of these proteins are gaining a lot of attention because they are likely to be involved in the cell-to-cell propagation of the pathology, and look more acutely toxic than large insoluble aggregates. For instance, in AD, oligomers of both Aβ and Tau have been shown to produce an immediate reduction of synaptic plasticity and memory when extracellularly applied (*Fá et al., 2016*). Intriguingly, the negative effects of oAβ and oTau occurred not only with high concentrations of Aβ or Tau alone, but also when sub-toxic doses of oAβ were combined with sub-toxic doses of oTau (*Fá et al., 2016*). These observations inspired the experiments shown in this manuscript. Here, we demonstrate that the suppression of APP, to which both oAβ and oTau can bind, causes a marked reduction of the oligomer entrance into neurons. Most importantly, we have found a common mechanism of action for extracellular Aβ and Tau oligomers, whose deleterious effect on LTP and memory depends upon the presence of endogenous APP.

Our finding that extracellular oAβ requires APP to impair synaptic plasticity and memory is consistent with previous studies showing that Aβ neurotoxicity might be mediated by APP, as suggested by the reduced vulnerability towards Aβ of cultured APP null neurons or mutated APP cells (*Lorenzo et al., 2000*; *Shaked et al., 2006*). This finding is also consistent with the observation that the presence of APP is likely to contribute to hippocampal hyperactivity, which has been suggested as a key mechanism of disease etiopathogenesis both in AD animal models and patients

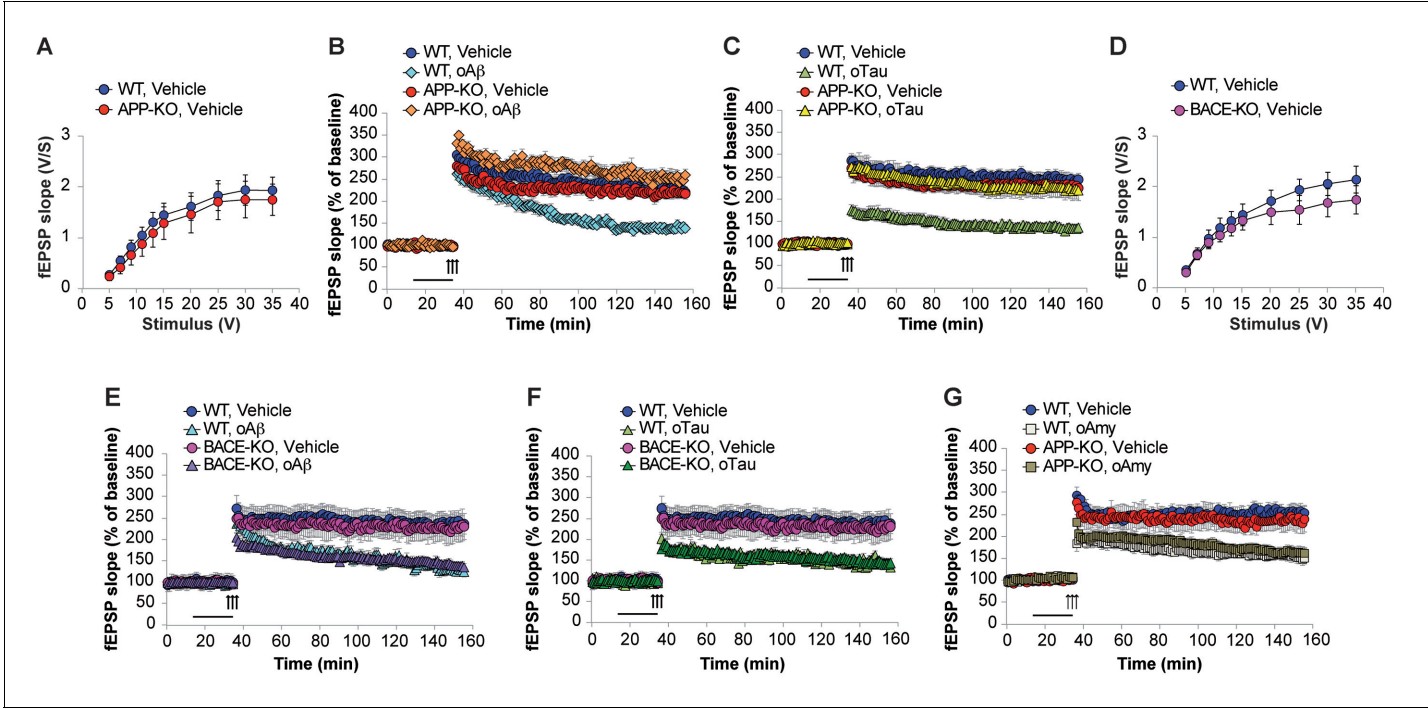

**Figure 5.** APP is necessary for extracellular oAβ and oTau to reduce LTP. (A) Basal synaptic transmission (BST) at the CA3-CA1 connection in slices from 3- to 4-month-old APP-KO mice was similar to WT littermates (n = 18 slices from WT vs. 18 slices from APP-KO; ANOVA for repeated measures $F_{(1,34)}$ = 0.416, p=0.524). (B) LTP was impaired in hippocampal slices from WT mice perfused with oAβ (200 nM), whereas there was no impairment in slices from APP-KO littermates. ANOVA for repeated measures $F_{(3,30)}$ = 19.738, p<0.0001. WT + vehicle vs. WT + oAβ: $F_{(1,16)}$ = 29.393, p<0.0001. WT + vehicle vs. APP-KO + oAβ: $F_{(1,13)}$ = 3.297, p=0.093. WT + vehicle: n = 9, WT + oAβ: n = 9, APP-KO + vehicle: n = 10, APP-KO + oAβ: n = 6. (C) LTP was impaired in hippocampal slices from WT mice perfused with oTau (100 nM), whereas there was no impairment in slices from APP-KO littermates. ANOVA for repeated measures $F_{(3,35)}$ = 11.033, p<0.0001. WT + vehicle vs. WT + oTau: $F_{(1,16)}$ = 50.543, p<0.0001. WT + vehicle vs. APP-KO + oTau: $F_{(1,16)}$ = 0.382, p=0.575. WT + vehicle: n = 8, WT + oTau: n = 10, APP-KO + vehicle: n = 11, APP-KO + oTau: n = 10. (D) CA3-CA1 BST in slices from 3- to 4-month-old BACE1-KO mice was similar to WT littermates (n = 24 slices from WT vs. 26 slices from BACE-KO; ANOVA for repeated measures $F_{(1,48)}$ = 0.714, p=0.402). (E) LTP was impaired in hippocampal slices from both WT and BACE-KO mice perfused with oAβ (200 nM). ANOVA for repeated measures $F_{(3,29)}$ = 5.738, p=0.003. WT + vehicle vs. WT + oAβ: $F_{(1,14)}$ = 23.663, p<0.0001. WT + vehicle vs. BACE-KO + oAβ: $F_{(1,14)}$ = 38.295, p<0.0001. WT + vehicle: n = 8, WT + oAβ: n = 8, BACE-KO + vehicle: n = 9, BACE-KO + oAβ: n = 8. F) LTP was impaired in hippocampal slices from both WT and BACE-KO mice perfused with oTau (100 nM). ANOVA for repeated measures $F_{(3,30)}$ = 6.919, p=0.001. WT + vehicle vs. WT + oTau: $F_{(1,14)}$ = 33.230, p<0.0001. WT + vehicle vs. BACE-KO + oTau: $F_{(1,15)}$ = 36.9961, p<0.0001. WT + oTau: n = 8, BACE-KO + oTau: n = 9. G) LTP was impaired in hippocampal slices from both WT and APP-KO mice perfused with oAmy (200 nM). ANOVA for repeated measures $F_{(3,38)}$ = 8.900, p<0.0001. WT + vehicle vs. WT + oAmy: $F_{(1,21)}$ = 34.694, p<0.0001. WT + vehicle vs. APP-KO + oAmy: $F_{(1,19)}$ = 19.277, p<0.0001. WT + vehicle: n = 11, WT + oAmy: n = 12, APP-KO + vehicle: n = 9, APP-KO + oAmy: n = 10.

The following source data is available for figure 5:

**Source data 1.** Data relating to *Figure 5A*.
**Source data 2.** Data relating to *Figure 5B*.
**Source data 3.** Data relating to *Figure 5C*.
**Source data 4.** Data relating to *Figure 5D*.
**Source data 5.** Data relating to *Figure 5E*.
**Source data 6.** Data relating to *Figure 5F*.
**Source data 7.** Data relating to *Figure 5G*.

(*Bakker et al., 2012*; *Busche et al., 2012*; *Palop et al., 2007*; *Verret et al., 2012*; *Vossel et al., 2013*). Along with these studies, APP has been demonstrated to bind Aβ monomers and dimers leading to activity-dependent APP-APP conformational changes that enhance neurotransmitter release (*Fogel et al., 2014*). When Aβ is accumulating in the brain, this increase of release probability might induce hippocampal hyperactivity resulting in failure of synaptic plasticity and memory loss (*Koppensteiner et al., 2016*).

Another interesting finding in our studies is that extracellular oTau requires APP to impair synaptic plasticity and memory. In support of this observation a few studies published several years ago, prior to the introduction of the concept of Tau oligomers, supported a direct interaction between APP and Tau (*Giaccone et al., 1996*; *Islam and Levy, 1997*; *Smith et al., 1995*). Moreover, recently, APP has been involved in the uptake of Tau fibrils into cells influencing Tau intracellular aggregation and spreading in the brain (*Takahashi et al., 2015*).

The dependence for the presence of APP shared by both oAβ and oTau in order to impair synaptic plasticity, suggests that APP is a key molecule involved in a common mechanism by which extracellular oAβ and oTau interfere with second messenger cascades relevant to memory formation. Indeed, Aβ and Tau share numerous biochemical characteristics and previous studies have suggested a possible common toxicity mechanism (*Gendreau et al., 2013*). Both peptides are β-sheet forming proteins, which explains their propensity for oligomerization and close association with membrane. Furthermore, both peptides can bind APP, a protein with structural similarities to type I transmembrane receptors, that might act as a cell surface receptor.

APP is also the precursor of Aβ (*Müller and Zheng, 2012*), which derives from sequential cleavage by γ- and β-secretases. We have therefore asked whether the toxicity of extracellular Aβ and Tau oligomers depends upon this amyloidogenic processing of APP. To this end we have used mice deficient in BACE1, the enzyme that initiates the amyloidogenic cascade. We found that BACE1-deficient mice are susceptible to the synapto-toxicity of oAβ and oTau in a similar fashion as WT littermates. Thus, amyloidogenic APP cleavage is not required for the impairment of LTP by the oligomers.

We also found that the APP dependence for the negative effect of oAβ and oTau onto LTP is specific to these proteins. This observation is consistent with the fact that both proteins are involved in AD. This conclusion derived from the experiments in which oAmy was capable of reducing LTP in APP-KO slices. Nevertheless, one cannot exclude that other β-sheet forming proteins besides Aβ and Tau require APP to impair synaptic plasticity. Regardless, the finding that Aβ and Tau share APP as a common mechanism for impairing LTP and memory is relevant and provides a common etiopathogenetic mechanism for their involvement in AD.

Our data are consistent with the hypothesis that APP serves as a common, direct molecular target for extracellular oAβ and oTau to impair LTP and memory. This is supported by the demonstration that both oAβ and oTau bind to APP. However, our experiments do not conclusively demonstrate that oligomer binding to APP is the cause of LTP and memory reduction, nor we can rigorously exclude the possibility that the two types of oligomers act on additional targets. For instance, it has been demonstrated that heparan sulfate and heparin sulphate proteoglycans bind with Aβ and Tau (*Holmes et al., 2013*; *Lindahl et al., 2009*) and mediate their internalization and neurotoxicity (*Holmes et al., 2013*; *Mirbaha et al., 2015*; *Sandwall et al., 2010*). Given that APP and heparan sulfate proteoglycans are likely to interact at the plasma membrane (*Reinhard et al., 2013*) and proteoglycans are rapidly degraded in the absence of proteins belonging to the APP superfamily (*Cappai et al., 2005*) proteoglycan degradation in the absence of APP might block the toxic action of oAβ and oTau. Nevertheless, the main observation of this manuscript showing that extracellular oAβ and oTau disrupt molecular mechanisms of synaptic plasticity and memory via APP is clear and has relevant implications for understanding AD etiopathogenesis.

We have found that suppression of APP reduces oAβ and oTau entrance into cells. This observation combined with the finding that intracellular perfusion with 6E10 antibodies recognizing the sequence 1–16 of human Aβ$_{42}$, rescues the LTP block by extracellular human oAβ (*Ripoli et al., 2014*), supports the hypothesis that, at least for oAβ, APP-dependent uploading of extracellular oligomers plays a critical role the impairment of synaptic plasticity, and presumably memory. APP might permit the entrance of the peptides into cells either directly into the cytosol or within vesicles during endocytosis, after which molecular mechanisms of learning and memory are impaired. A direct entrance into the cytosol might occur if APP functions as a channel through which the two

oligomers both with a diameter in the low nm range (*Cizas et al., 2010*; *Fá et al., 2016*) enter cells. In agreement with this hypothesis, it has been reported that APP forms a non-selective channel when injected in *Xenopus* oocytes (*Fraser et al., 1996*). A variant on this hypothesis is that APP permits the formation of pores/channels by the oligomers, as ion conductance across lipid bilayers is increased by oligomers of several different amyloids (*Kayed et al., 2004*), which affect the permeability of the plasma membrane, leading to elevation of intracellular [Ca$^{2+}$] and toxic changes. With this regard, A$\beta$ has been reported to form large conductance, non-specific ion channels (*Fraser et al., 1997*). The endocytotic mechanism, in turn, is supported by the demonstration that full-length APP is a transmembrane protein, which is endocytosed from the cell surface into endosomes (*Nordstedt et al., 1993*) and by studies showing endocytosis of Tau (*Wu et al. 2013*).

Another aspect of our experiments is that APP suppression does not appear to dramatically affect the number of MAP2-positive cells taking up A$\beta$ (only ~80% of WT cells), but does reduce the number of intracellular aggregates for the peptide per neuron (~55% of WT cells), whereas APP suppression clearly affects both the number of cells taking up Tau (~50%) and the number of intracellular aggregates for oTau (~52%). These findings open the question of whether the number of cells taking up oligomers and the amount of intracellular aggregates may reflect two different processes, i.e., uptake and degradation/clearance of aggregates. This is an interesting possibility that might explain our observations. Of note, to date there is no data showing that APP is involved in modulation of A$\beta$/Tau clearance. Nevertheless, it would be interesting in future experiments to explore whether mechanisms controlling protein degradation/clearance in neurons are regulated by APP.

Albeit the experiments on oligomer entrance support the hypothesis that APP serves as a Trojan horse for oA$\beta$ and oTau to enter neurons prior to impairing second messenger cascades relevant to synaptic plasticity and memory formation, they do not exclude an alternative scenario in which oligomer interaction with APP activates the intracellular segment of APP, AID/AICD, triggering a cascade of events leading to derangement of memory mechanisms. In support of this hypothesis, it has been shown that phosphorylation of the intracellular threonine 668 of APP mediates synaptic plasticity deficits and memory loss (*Lombino et al., 2013*). Moreover, the AID/AICD fragment of APP could form a multimeric complex with the nuclear adaptor protein Fe65 and the histone acetyltransferase Tip60, potentially stimulating transcription (*Cao and Südhof, 2001*). If so, the entrance of the oligomers might serve other purposes rather than impairing synaptic plasticity and memory. For instance, it has been suggested that Tau entrance leads to propagation of Tau misfolding (*Frost et al., 2009*). Nevertheless, our findings that APP is necessary for impairment of LTP and memory following elevation of A$\beta$ and Tau is still relevant, as it sheds light into how the oligomers cause memory loss in AD and other neurodegenerative disorders.

The prevailing hypothesis in the AD field is that A$\beta$ triggers Tau pathology. Our data, however, do not support this hypothesis in which A$\beta$ and Tau are placed in series but suggest a different scenario in which extracellular A$\beta$ and Tau oligomers act in parallel, both through APP. Interestingly, this hypothesis would also explain why tauopathies result in neuronal loss similar to AD but in the absence of A$\beta$. The identification of the common biochemical neuronal modifications occurring after the APP involvement and underlying the derangement of the molecular mechanisms of gene transcription involved in memory formation, is beyond the scope of the present manuscript. Nevertheless, our findings are translationally significant, as they have permitted the identification of a common molecule, APP, which might be therapeutically targeted at sites serving for direct interaction with A$\beta$ and Tau oligomers or, alternatively, with proteins downstream of such oligomers, other than the classical $\beta$- and $\gamma$-secretase sites.

## Materials and methods

### Animals

All protocols involving animals were approved by Columbia University (#AC-AAAO5301), Università di Catania (#327/2013-B, #119–2017-PR), Università Cattolica del Sacro Cuore (#626–2016-PR), Albert Einstein College of Medicine (#20160407), and the respective Institutional Animal care and Use Committee (IACUC); experiments involving animals were performed in accordance with the relevant approved guidelines and regulations. C57BL/6J (RRID:IMSR_JAX:000664) and *App*-KO (Jackson Lab B6.129S7-Apptm1Dbo/J; RRID:IMSR_JAX:004133) mice and their littermates were obtained

from breeding colonies kept in the animal facility of Columbia University, Università di Catania, and Università Cattolica del Sacro Cuore. *Bace1*-KO mice (*Luo et al., 2001*) and their WT littermates were obtained from a breeding colony kept at Albert Einstein College of Medicine which derived from mice that were originally donated by Dr. Vassar at Northwestern University. They were 3–4 months of age except newborn mice for cell cultures. Both sexes were used. All mice were maintained on a 12 hr light/dark cycle (lights on at 6:00 AM) in temperature and humidity-controlled rooms; food and water were available ad libitum.

## Oligomer preparation

### Tau oligomers

Human Tau preparation and oligomerization was obtained as described previously with slight modifications (*Fá et al., 2016*). Briefly, a recombinant Tau 4R/2N construct containing C-terminal 6x His-tag was transfected in Escherichia coli (Rosetta). Cells were streaked on LB agar ampicillin plates and a single colony was picked and grown overnight in a shaker at 37°C in 100 ml Expansion Broth (Zymo Research) and 300 ml Overexpression Broth (Zymo Research). Cells were pelleted at 4 °C by centrifugation at 6000 × g. After a freeze-thaw cycle, cells were lysed in a 2% Triton X-100 PBS and with a protease inhibitor mixture (Complete, EDTA-free; Roche Diagnostics). Streptomycin sulfate was added to precipitate DNA. After incubation for 5 min at 4°C followed by sonication, the preparation was heated at 100°C for 15 min, and centrifuged to remove the precipitate. TCEP-HCl (ThermoScientific) and 1% perchloric acid were added to the supernatant prior to neutralizing it with 1N NaOH, and placing it in a Pierce protein concentrator (PES, 30K MWCO) (ThermoScientific) to be centrifuged at 4000 × g. The resulting supernatant was loaded on His-Spin Protein Miniprep columns (Zymo Res.) and eluted with phosphate buffer containing 300 mM NaCl plus 250 mM imidazole. Eluted tau was then treated with TCEP-HCl and EDTA, and incubated at room temperature (RT) for 1 hr. Oligomerization buffer was next added to the treated eluted Tau prior to centrifuging it in a PES at 4000 × g. Oligomerization was achieved via introduction of disulfide bonds through incubation with 1 mM $H_2O_2$ at room RT for 20 hr, followed by centrifugation in a PES at 4000 × g. The resulting material was used for our experiments. Tau protein concentration was determined from the absorption at 280 nm with an extinction coefficient of 7450 $cm^{-1}$ $M^{-1}$.

### Aβ oligomers

Human $A\beta_{42}$ oligomerization was obtained as described previously (*Puzzo et al., 2005*; *Watterson et al., 2013*). Briefly, a protein film was prepared by dissolving $A\beta_{42}$ lyophilized powder (Biopolymer Laboratory, UCLA, CA, USA or American Peptide, CA, USA) in 1,1,1,3,3,3-Hexafluoro-2-Propanol (HFIP) and subsequent incubation for 2 hr at RT to allow complete monomerization. The Aβ film was dissolved in dimethylsulfoxide (DMSO), sonicated for 15 min, aliquoted, and stored at −20°C. To oligomerize the peptide, phosphate buffered saline (PBS) was added to an aliquot of DMSO-Aβ to obtain a 5 mM solution that was incubated for 12 hr at 4°C. This oligomerized Aβ solution was then diluted to the final concentration of 200 nM in artificial cerebrospinal fluid (ACSF) composed as following: 124.0 NaCl, 4.4 KCl, 1.0 $Na_2HPO_4$, 25.0 $NaHCO_3$, 2.0 $CaCl_2$, 2.0 $MgCl_2$ in mM.

### Amylin oligomers

Human Amy oligomerization was obtained as described previously (*Ripoli et al., 2014*). Briefly, a protein film was prepared by dissolving Amy lyophilized powder (Anaspec, CA, USA) in HFIP and subsequent incubation for 2 hr at RT to allow complete monomerization. The Amy film was dissolved in DMSO, sonicated for 15 min, aliquoted, and stored at −20°C. To oligomerize the peptide, PBS was added to an aliquot of DMSO-Amy to obtain a 5 mM solution that was incubated for 12 hr at 4°C. This oligomerized Amy solution was then diluted to the final concentration of 200 nM in ACSF.

## Co-Immunoprecipitation

WT and APP695 with the Swedish mutation (APPSw) overexpressing human embryonic kidney (HEK293; RRID:CVCL_0045) cells were used to examine the molecular interaction between Tau oligomers and APP. HEK293 cells were originally obtained from ATCC and verification of the cell line was validated by STR profiling (see; https://www.atcc.org/Products/All/CRL-1573.aspx#specifications). Testing for potential mycoplasma was performed using Hoechst 33258 as a marker for

indirect DNA fluorescent staining (protocol described at: http://www.sigmaaldrich.com/technical-documents/protocols/biology/testing-for-mycoplasma0.html).

APPSw and untransfected cells were maintained in DMEM supplemented with 10% fetal bovine serum. Membrane fractions were prepared by homogenizing cells in buffer (5 mM HEPES pH 7.4, 1 mM EDTA, 0.25 M sucrose, protease inhibitor cocktail). Extracts were clarified by centrifugation (1000 × g, 5 min, 4°C) and membrane fractions were obtained by centrifuging supernatant (100,000 × g, 1 hr, 4°C). Membranes from control (endogenous APP only) and APPSw expressing cells were solubilized under mild conditions (25 mM HEPES pH 7.4, 150 mM NaCl, 2 mM EDTA, 1% CHAPSO, protease inhibitor cocktail), diluted to 0.5% CHAPSO - to maintain normal lipidation of APP and native protein conformation - and incubated with Tau 4R/2N oligomers (10 µg, 3–4 hr, 4°C). Samples (1.11 mg total protein) were incubated with a monoclonal antibody directed to the APP C-terminal domain (C1/6.1 *Mathews et al., 2002*); 5 µg, 2 hr) and immunoprecipitated using Protein G-Sepharose. Non-specific bound proteins were removed by successive washing with lysis buffer. Immunoprecipitated APP complexes were eluted with Laemmli buffer, resolved by SDS-PAGE (4–12% Bis-Tris gels, BioRad) and probed for Tau using Tau-5 antibodies (1:1,000) and APP-CTF (C1/6.1) to confirm the immunoprecipitation efficiency as well as the interaction.

## Far western blotting (fWB)

APP-Tau interaction was detected performing fWB as previously described (*Wu et al., 2007*). Hippocampal neurons were lysed in cold RIPA buffer containing 1 mM phenylmethylsulfonyl fluoride, phosphatase and protease inhibitor mixtures (Sigma) and 0.1% sodium dodecyl sulfate. After incubation for 30 min on ice and centrifugation (10,000 × g for 30 min at 4°C), the supernatant was removed and protein concentration was determined using the Bio-Rad protein assay. Each protein sample (30 µg) was separated on 8% SDS-polyacrylamide gel and blotted onto nitrocellulose membranes (Millipore Co., Bedford, MA). The blotted proteins were then denatured with guanidine–HCl and then renatured by gradually reducing guanidine concentration (from 6 to 0 M). The last renaturing step with the guanidine-HCl-free buffer was maintained overnight at 4°C. The membrane was blocked with 5% nonfat dry milk in TBS, 1% tween-20 (TBST) for 1 hr at RT and then incubated overnight at 4°C with 10 µg of purified 'bait' protein oTau. After three washes with TBST, the membrane was incubated with one of the following primary antibodies for 1 hr a RT in 3% milk in the TBST buffer: anti Tau Ab-2 (clone Tau-5, Thermo Fisher Scientific, Waltham, MA). Membranes were then stripped by heating at 56°C in 62.5 mM Tris-HCl, pH 6.7, with 100 mM 2-mercaptoethanol and 2% SDS and re-incubated with anti APP-C terminus (Sigma) to check whether Tau and APP are on the same position on the membrane. Blots were developed with the Pierce ECL Plus Western Blotting Substrate (Thermo Fisher Scientific) and visualized using UVItec Cambridge Alliance. The BenchMark Pre-Stained Protein Ladder (Invitrogen) was used as molecular mass standards.

## Assessment of oligomers entrance into neurons

### Primary hippocampal neuronal cultures

Primary cultures of hippocampal neurons were obtained from C57BL/6J mice (WT), and B6.129S7-Apptm1Dbo/J (APP KO) mice as previously described (*Piacentini et al., 2015*; *Scala et al., 2015*). Briefly, hippocampi dissected from the brain of E18 mice embryos were incubated for 10 min at 37°C in PBS containing trypsin-ethylenediaminetetraacetic acid (0.025%/0.01% wt/vol; Biochrom AG, Berlin, Germany), and the tissue was then mechanically dissociated at RT with a fire-polished Pasteur pipette. The cell suspension was harvested and centrifuged at 235 × g for 8 min. The pellet was suspended in 88.8% Minimum Essential Medium (Biochrom), 5% fetal bovine serum, 5% horse serum, 1% glutamine (2 mM), 1% penicillin-streptomycin-neomycin antibiotic mixture (Invitrogen, Carlsbad, CA), and glucose (25 mM). Cells were plated at a density of $10^5$ cells on 20 mm coverslips (for immunocytochemical studies) and $10^6$ cells/well on 35 mm six-well plates (for fWB studies), precoated with poly-L-lysine (0.1 mg/mL; Sigma, St. Louis, MO). Twenty-four hours later, the culture medium was replaced with a mixture of 96.5% Neurobasal medium (Invitrogen), 2% B-27 (Invitrogen), 0.5% glutamine (2 mM), and 1% penicillin streptomycin- neomycin antibiotic mixture. After 72 hr, this medium was replaced with a glutamine free version of the same medium, and the cells were grown for 10 more days before experiments.

## Preparation of labeled Aβ

Freeze-dried human synthetic Aβ$_{42}$ labeled with HiLyteTM Fluor 555 at the C-terminus (oAβ−555) was purchased from AnaSpec (Fremont, CA). Protein solution was prepared as previously described (Ripoli et al., 2013). Briefly, peptide was diluted to 1 mM in 1,1,1,3,3,3,-hexafluoro-2-propanol to disassemble preformed aggregates and stored as dry films at −20°C before use. The film was dissolved at 1 mM in DMSO, sonicated for 10 min, diluted to 100 μM in cold PBS, and incubated for 12 hr at 4°C to promote protein oligomerization. Aβ$_{42}$-555-labeled preparation was purified with Amicon Ultra Centrifugal Filter (2 KDa) and then was resuspended in PBS at a concentration of 100 mM before final dilution in the culture medium.

## Preparation of labeled Tau

oTau preparations were labeled with the IRIS-5-NHS active ester dye (IRIS-5; λex: 633 nm; λem: 650–700 nm; Cyanine Technology Turin, Italy) as previously described (Fá et al., 2016). Briefly, Tau solutions (2 μM in PBS) were mixed with 6 mM IRIS-5 in DMSO for 4 hr in the dark under mild shaking conditions. After this time, labeled Tau was purified with Vivacon 500 ultrafiltration spin columns (Sartorius Stedim Biotech GmbH) and then resuspended in PBS and used at final concentration of 100 nM in the culture medium.

## Assessment of oAβ and oTau entrance into neurons

WT and APP KO hippocampal neurons at 14 days in vitro were treated with either 200 nM Aβ$_{42}$-555 or 100 nM oTau-IRIS5 for 20 min. After treatment cells were fixed with 4% paraformaldehyde in PBS for 10 min at RT, permeabilized for 15 min with 0.3% Triton X-100 [Sigma] in PBS, and incubated for 30 min with 0.3% BSA in PBS to block nonspecific binding sites. The primary antibody rabbit anti microtubule associated protein 2 (MAP2, Immunological Sciences, Rome, Italy; 1:200 overnight at 4°C for 90 min) and the corresponding secondary antibody Alexa Fluor 488 donkey anti-rabbit (Invitrogen 1:1000 for 90 min at RT) were then used to recognize neurons. Nuclei were counterstained with 4′,6-diamidino-2-phenylindole (DAPI, 0.5 μg/ml for 10 min; Invitrogen) and cells were coverslipped with ProLong Gold anti-fade reagent (Invitrogen), before being studied through high-resolution confocal microscopy. Confocal stacks made of images (512 × 512 pixels) were acquired with a confocal laser scanning system (Nikon A1 MP) and an oil-immersion objective (60× magnification; N. A. 1.4). Additional 2.5× magnification was applied to obtain a pixel size of 90 nm. Fluorescent dyes were excited with diode lasers (405, 488, 546 and 633 nm). The following criteria were used for spot detection: XY area ≥200 nm$^2$; Z height ≥1.5 μm. The studied proteins were considered internalized when the overlapping of MAP2 fluorescence with the height of the fluorescent spots was greater than 65% (~1 μm). Conversely, they were considered attached to the neuronal surface when the fluorescence signals were close to each other, but with less than 30% overlap. Spots internalized in neurons were detected and counted by the Image J software, through an algorithm that automatically detects co-localization between MAP2 fluorescence and either Aβ555 or Tau-IRIS5. MAP2 fluorescence was binarized to form a mask of the fluorescence pattern for every single XY plane of the Z stacks, and this mask was multiplied plane-by-plane for the corresponding fluorescence of Aβ or Tau stacks. This operation selected only Aβ and Tau signals associated with MAP2-positive areas by deleting any Aβ and Tau signals unrelated to MAP-2 immunoreactivity. The resulting fluorescence signals gave an unbiased estimate of Aβ or Tau oligomers internalized in neurons within each microscopic field. The number of fluorescent spots were then counted by the 'analyze particle' macro of Image J after having done a maximum intensity projection of every Z stacks. To provide a global estimate of the protein uploading into neurons, internalization of oAβ and oTau was also quantified through the 'internalization index' obtained by multiplying the percentage of neurons internalizing fluorescent proteins by the mean number of fluorescent spots inside neurons. Assessment of fluorescent protein oligomers attached to the neuronal surface was carried out by spanning the XZ-YZ planes from Z stacks for every microscopic field acquired.

## Behavioral studies

### Intrahippocampal administration of oAβ and oTau

To perform intrahippocampal infusions of oligomers, mice underwent stereotaxic surgery for cannulas implantation. After anesthesia with Avertin (500 mg/Kg), mice were implanted with a 26-gauge

guide cannula into the dorsal part of the hippocampi (coordinates from *bregma*: posterior = 2.46 mm, lateral = 1.50 mm to a depth of 1.30 mm). After 6–8 days of recovery, mice were bilaterally infused with oA$\beta$ or oTau preparations or vehicle in a final volume of 1 µl over 1 min with a microsyringe connected to the cannulas via polyethylene tubing. During infusion, animals were handled gently to minimize stress. After infusion, the needle was left in place for another minute to allow diffusion. In some animals, after behavioral studies, a solution of 4% methylene blue was infused for localization of infusion cannulas.

## Fear conditioning

Fear conditioning was performed as previously described (*Fiorito et al., 2013*; *Watterson et al., 2013*). Our conditioning chamber, equipped with a camera placed on the top of the cage, was in a sound-attenuating box. The conditioning chamber had a bar insulated shock grid floor, removable. After each experimental test the floor was cleaned with 75% ethanol. Mice were handled once a day for 3 days before behavioral experiments. Only one animal at a time was present in the experimentation room. During the first day, mice were placed in the conditioning chamber for 2 min before the onset of a discrete tone [conditioned stimulus (CS)] (a sound that lasted 30 s at 2800 Hz and 85 dB). In the last 2 s of the CS, mice were given a foot shock [unconditioned stimulus (US)] of 0.80 mA for 2 s through the bars of the floor. After the CS/US pairing, the mice were left in the conditioning chamber for 30 s and then they were placed back in their home cages. Freezing behavior, defined as the absence of all movement except for that necessitated by breathing, was manually scored. During the second day, we evaluated the contextual fear learning. Mice were placed in the conditioning chamber and freezing was measured for five consecutive min. During the third day, we evaluated the cued fear learning. Mice were placed in a novel context (rectangular black cage with vanilla odorant) for 2 min (pre-CS test), after which they were exposed to the CS for 3 min (CS test), and freezing was measured. Sensory perception of the shock was determined 24 hr after the cued test through threshold assessment. Foot shock intensity started at 0.1 mA and increased by 0.1 mA every 30 s. We recorded the first visible, motor and vocal response.

## 2 day radial arm water maze (RAWM)

RAWM was performed as previously described (*Watterson et al., 2013*). During the first day, mice were trained in 15 trials to identify the platform location in a goal arm by alternating between a visible and a hidden platform from trial 1 to 12 (beginning with the visible platform in the assigned arm). In the last four trials (trial 13–15) only a hidden platform was utilized. During the second day the same procedure was repeated by using only the hidden platform from trial 1 to 15. An entrance into an arm with no platform, or failure to select an arm after 15 s was counted as an error and the mouse was gently pulled back to the start arm. The duration of each trial was up to 1 min. At the end of each trial, mouse rested on the platform for 15 s. The goal arm was kept constant for all trials, with a different starting arm on successive trials. Data were analyzed and displayed as averages of blocks of 3 trials per mouse. A visible platform test was performed to control for possible motivational, visual and motor deficits. This consisted in a two-day test, with two sessions/day (each consisting of three 1 min trials), in which we recorded the time taken to reach a visible platform (randomly positioned in a different place each time) marked with a green flag.

## Open field

Open Field was performed as previously described (*Fá et al., 2016*). Our arena consisted in a white plastic bow divided into sectors (periphery and center) by black lines. Each mouse started the test randomly from one of the border, and was permitted to freely explore the arena for 5 min in two consecutive days. The test was performed in a quiet, darkened room and one light bulb provided a bright illumination. We scored the % time spent into the center and the number of entries into the center.

## Electrophysiological recordings

Electrophysiological recordings were performed as previously described (*Puzzo et al., 2005*). Briefly, transverse hippocampal slices (400 µm) were cut and transferred to a recording chamber where they were maintained at 29°C and perfused with ACSF (flow rate 2 ml/min) continuously bubbled with

95% $O_2$ and 5% $CO_2$. Field extracellular recordings were performed by stimulating the Schaeffer collateral fibers through a bipolar tungsten electrode and recording in CA1 *stratum radiatum* with a glass pipette filled with ACSF. After evaluation of basal synaptic transmission, a 15 min baseline was recorded every minute at an intensity eliciting a response approximately 35% of the maximum evoked response. LTP was induced through a theta-burst stimulation (4 pulses at 100 Hz, with the bursts repeated at 5 Hz and three tetani of 10-burst trains administered at 15 s intervals). Responses were recorded for 2 hr after tetanization and measured as field excitatory post-synaptic potentials (fEPSP) slope expressed as percentage of baseline.

## Statistical analyses

All experiments were in blind with respect to treatment. All data were expressed as mean ± standard error mean (SEM). For experiments on oligomer entrance into cultured neurons pairwise comparisons were performed through Student's *t* test. Behavioral experiments were designed in a balanced fashion and, for each condition mice were trained and tested in three to four separate sets of experiments. Freezing, latency, time spent in the center of the arena and number of entries in the center were manually scored by an expert operator by using a video-tracking recording system. We used one-way ANOVA with Bonferroni post-hoc correction or ANOVA with repeated measures for comparisons among the four groups of mice. For electrophysiological recordings on slices, results were analyzed in pClamp 10 (Molecular Devices; RRID:SCR_011323) and compared by ANOVA with repeated measures considering 120 min of recording after tetanus or the 26th-30th recording points. Statistical analysis was performed by using Systat 9 software (Chicago, IL, USA; RRID:SCR_010455). For protein entrance into neurons we used Student's *t*-test to compare the internalization index between WT and APP KO neurons. The level of significance was set at $p < 0.05$.

## Acknowledgements

This work was supported by NIH grants R01AG049402 (OA), Italian FFO (DP and AP), Canadian Institute of Health Research TAD-117950 (PEF), Catholic University intramural funds (CG).

## Additional information

### Funding

| Funder | Grant reference number | Author |
|---|---|---|
| National Institutes of Health | R01AG049402 | Ottavio Arancio |
| Italian FFO | | Daniela Puzzo Agostino Palmeri |
| Canadian Institutes of Health Research | | Paul Fraser |
| Catholic University Intramural Funds | | Claudio Grassi |

The funders had no role in study design, data collection and interpretation, or the decision to submit the work for publication.

### Author contributions

DP, Conceptualization, Resources, Data curation, Formal analysis, Investigation, Methodology, Writing—original draft, Writing—review and editing; RP, Data curation, Formal analysis, Investigation, Methodology, Writing—review and editing; MF, Investigation, Methodology, Writing—review and editing; WG, DDLP, AS, HZ, MRT, SC, Investigation, Methodology; AP, PF, Resources, Writing—review and editing; LD, CG, Conceptualization, Resources, Writing—review and editing; OA, Conceptualization, Resources, Supervision, Funding acquisition, Validation, Writing—original draft, Project administration, Writing—review and editing

## Author ORCIDs

Daniela Puzzo, http://orcid.org/0000-0002-9542-2251
Claudio Grassi, http://orcid.org/0000-0001-7253-1685
Ottavio Arancio, http://orcid.org/0000-0001-6335-164X

## Ethics

Animal experimentation: This study was performed in strict accordance with the recommendations in the Guide for the Care and Use of Laboratory Animals of the National Institutes of Health and the European Community Council. All protocols involving animals were approved by Columbia University (#AC-AAAO5301), Università di Catania (#327/2013-B, #119-2017-PR), Università Cattolica del Sacro Cuore (#626-2016-PR), Albert Einstein College of Medicine (#20160407), and the respective Institutional Animal care and Use Committee (IACUC).

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
