## [Decision Letter]

Thank you for submitting your article "LTP and memory impairment caused by extracellular Aβ and Tau oligomers is APP-dependent" for consideration by *eLife*. Your article has been favorably evaluated by a Senior Editor and two reviewers, one of whom is a member of our Board of Reviewing Editors. The reviewers have opted to remain anonymous.

The reviewers have discussed the reviews with one another and the Reviewing Editor has drafted this decision to help you prepare a revised submission.

General assessment and central conclusions: The reviewers find that the manuscript describes and interesting set of studies demonstrating that both Aβ oligomers and Tau oligomers require endogenous APP to impair behavior (contextual fear and radial arm water maze) and LTP. This is particularly important because it suggests a common therapeutic target that could impact both Aβ and Tau-mediated cell dysfunction. The authors also include data that amylin oligomers do not require APP to impair LTP, strongly suggesting that Aβ and Tau oligomers are specifically acting to cause synaptic dysfunction, as opposed to a general effect by aggregated proteins.

The reviewers raise a number of concerns that must be addressed before the paper can be accepted.

Essential revisions:

1) For Figure 2 there was concern that the uptake of oligomers into neurons was not convincing and based on a small number of cells. We suggest rather than counting manually that an unbiased imaging approach be used that would allow a larger number of cells to be counted and eliminate and counting bias. Straight numbers rather than the internalization index would be better if larger numbers of cells are counted. Please also address whether cell surface oligomers differ between genotypes and whether the patterns from Aβ and Tau are similar or different.

2) The APP-KO result is very interesting and would be strengthened from a mechanistic perspective by the inclusion of data using BACE and/or γ secretase. Is APP processing required for Aβ and Tau toxicity?

---

## [Author Response]

Essential revisions:

1) For Figure 2 there was concern that the uptake of oligomers into neurons was not convincing and based on a small number of cells. We suggest rather than counting manually that an unbiased imaging approach be used that would allow a larger number of cells to be counted and eliminate and counting bias.

To address this critique, we performed additional experiments to increase the number of analyzed cells, now ranging from n = 84 to n = 127 in the different experimental groups. As requested, unbiased counts of Aβ and Tau fluorescent spots were performed by an algorithm in ImageJ software. The new data are reported in the subsection “Expression of APP is required for efficient intra-neuronal uptake of oAβ and oTau” and in the revised Figure 2. Methodological details are described in the subsection “Assessment of oAβ and oTau entrance into neurons”.

Straight numbers rather than the internalization index would be better if larger numbers of cells are counted. Please also address whether cell surface oligomers differ between genotypes and whether the patterns from Aβ and Tau are similar or different.

In the revised Figure 2 we added panels showing the percentage of cells that internalized Aβ and Tau as well as the numbers of internalized Aβ and Tau spots.

2) The APP-KO result is very interesting and would be strengthened from a mechanistic perspective by the inclusion of data using BACE and/or γ secretase. Is APP processing required for Aβ and Tau toxicity?

We have performed this study by using BACE-KO mice. We found that these animals did not confer resistance to the damage of LTP by Aβ and Tau oligomers, demonstrating that APP processing is not required for Aβ and Tau toxicity (see Results subsection “APP is necessary for extracellular oAβ and oTau to reduce LTP”, second paragraph; Discussion, fifth paragraph and Figure 5).